# A Pilot Study of Methods for Evaluating the Effects of Arousal and Emotional Valence on Performance of Racing Greyhounds

**DOI:** 10.3390/ani10061037

**Published:** 2020-06-15

**Authors:** Melissa Starling, Anthony Spurrett, Paul McGreevy

**Affiliations:** Sydney School of Veterinary Science, School of Life and Environmental Sciences, Faculty of Science, University of Sydney, Camperdown, NSW 2006, Australia; aspu5654@uni.sydney.edu.au (A.S.); paul.mcgreevy@sydney.edu.au (P.M.)

**Keywords:** dog, eye temperature, infrared thermography

## Abstract

**Simple Summary:**

Racing greyhounds in Australia may have their racing careers ended early if they do not reliably chase the lure. It is not known why greyhounds bred specifically to chase may fail to do so, but possible reasons may be they are overly distressed by the race meet environment, or they get frustrated that they are not able to capture the lure. The current study sought ways to investigate these possibilities by exploring potential indicators of arousal and frustration in greyhounds at race meets across three different racetracks in New South Wales (NSW). Eye temperature offers a non-invasive, objective way to assess arousal. Behaviour thought to indicate arousal was also recorded. Finally, behaviour was recorded at the end of races to seek indications of frustration. Greyhounds that were older and had higher eye temperatures after the race performed worse in their races. Eye temperature before races was affected by the racetrack at which the race meet was held as well as the race number, suggesting some tracks may be more stressful for greyhounds than others, and that kennelling is probably a source of ongoing distress at race meets. Frustration-related behaviours were common at two of the tracks but less common at the third, where toys were available in the catch pen.

**Abstract:**

The racing greyhound industry in Australia has come under scrutiny in recent years due to animal welfare concerns, including wastage where physically sound greyhounds fail to enter or are removed from the racing industry because of poor performance. The reasons why some greyhounds perform poorly in racing are not well understood, but may include insufficient reinforcement for racing or negative affective states in response to the race meet environment. The current study investigated ways to measure affective states of greyhounds (*n* = 525) at race meets across three racetracks and the factors influencing performance by collecting behavioural and demographic data, and infrared thermographic images of greyhounds’ eyes at race meets. Increasing Eye Temp After had a negative association with performance (*n* = 290, Effect = −0.173, s.e. = 0.074, *p*-value = 0.027), as did increasing age (*n* = 290, Effect = −0.395, s.e. = 0.136, *p*-value = 0.004). The start box number also had a significant association, with boxes 4, 5 and 7 having an inverse relationship with performance. There was a significant effect of racetrack on mean eye temperatures before and after the race (*n* = 442, Effect = 1.910, s.e. = 0.274, *p*-value < 0.001; Effect = 1.595, s.e. = 0.1221, *p*-value < 0.001 for Gosford and Wentworth respectively), suggesting that some tracks may be inherently more stressful for greyhounds than others. Mean eye temperature before the race increased as the race meet progressed (*n* = 442, Effect = 0.103, s.e. = 0.002, *p*-value < 0.001). Behaviours that may indicate frustration in the catching pen were extremely common at two of the tracks but much less common at the third, where toys attached to bungees were used to draw greyhounds into the catching pen.

## 1. Introduction

Greyhound racing in Australia is an activity supported largely by revenue from betting on the outcome of races. As such, there may be pressure on owners of racing greyhounds to race their dogs as often as the dogs are physically able so that race winnings support the greyhounds’ upkeep and further racing activities. Recent scrutiny into the greyhound racing industry in Australia, and in particular, the state of New South Wales (NSW), has raised questions about the level of so-called wastage of greyhounds within the racing industry. Wastage describes the process by which greyhounds fail to enter racing or are discarded from racing because they do not perform the task they were bred for, i.e., racing [1]. It can be physical e.g., from lameness, or behavioural e.g., from relative disinterest in running. The ultimate fate of discarded dogs is unknown but may include rehoming to a companion home, being retired but retained by the original owner or trainer, or being euthanised. Failing to chase a lure along with marring (interfering with another dog on the track during a race) attracts penalties of temporary race bans. Failure to chase is considered a form of behavioural wastage, which is where otherwise physically healthy and sound animals are removed from a role because they are unable to perform it adequately due to behavioural unsuitability [2]. Greyhounds were not tracked at the time of data collection, so no figures are available on the magnitude of behavioural wastage in greyhound racing in NSW. It is thought to account for as much as 50%–70% of the greyhounds lost from racing [3], although this may be an over-estimation [2], and does not differentiate between behavioural wastage and greyhounds that have suffered injuries. This is why it is important to understand why greyhounds may fail to adequately perform the activity they were specifically bred to perform. Behavioural wastage is a multi-faceted issue with many potential contributing factors, and little research has been conducted on it to date. Although it is likely that most behavioural wastage takes place before greyhounds reach the track [1], a key component of understanding why greyhounds may fail to chase lies in understanding their experience of race meets.

Greyhound races in Australia begin with a so-called stir-up, which involves greyhounds being given an opportunity to watch the lure traverse the track; usually twice while they are in a pen outside the track. Under Australian greyhound racing rules, pre-stir-up occurs approximately ten minutes before the start of each race, and five minutes before the stir-up [4]. Pre-stir-up involves collecting the dogs from their kennel, walking them to a grass area next to the track where they may eliminate, and fitting them with a racing rug and any additional pre-race preparations such as taping body parts for protection against injury. After being allowed to witness the lure traverse the track, the greyhounds are walked to their starting boxes, loaded into the boxes, and then released from the boxes to chase the lure for a set distance. At the end of the race, a gate is swung across the track immediately behind the lure to stop the greyhounds from chasing it. The lure itself draws away from the greyhounds and passes through a small flap in this gate, and the greyhounds are diverted into the catching pen alongside the track, where they are caught by handlers and then led from the track. The catching pen is unique to Australia and, it is argued by industry participants [5], may be a source of frustration for greyhounds, firstly because they are unable to capture the lure and, secondly, because there is rarely an object in the catching pen for them to interact with in lieu of the lure. The consequences of frustration may be a subsequent failure to chase or redirection of frustration onto nearby dogs, both of which attract penalties if they occur during races (but not in the catching pen). The risk of injury in the catching pen may increase if greyhounds redirect frustration onto nearby conspecifics as race participants are decelerating at different rates.

Many factors influence the performance of racing animals. Previous research on racehorses has shown that horses that finished as losers (bottom 20% of finishers) showed behaviour in the mounting yard associated with high arousal immediately before the race [6]. High arousal before the race may lead to a reduction in fine motor control [7], and may also compromise judgement and cognitive processing [8], or indicate an animal expending energy required for racing on active behaviours before the race.

Heightened arousal in greyhounds prior to racing may be caused by distress related to the racetrack environment including kennelling. Anticipation may also heighten arousal levels [9,10,11], which may, in turn, be influenced by how long the dog has been kennelled at the racetrack, or how many days it has been since the dog last raced, or how experienced the dog is with the procedure at race meets. A previous study on racing greyhounds revealed an increase in arousal in dogs that raced as well as those that merely watched racing [12], suggesting greyhound arousal increases with anticipation of an opportunity to race. Road transport over an hour in duration is regarded as distressing for livestock animals [13], and studies of dogs undergoing air travel show an associated increase in behavioural and physiological signs of distress [14,15]. Therefore, travel time to the race meet may also influence pre-race arousal.

Infrared thermographic (IRT) cameras are increasingly being used to record the surface temperature of non-human animals’ eyes. Increases in eye temperature detected by IRT have revealed arousal in a variety of animals including mice [16], rabbits [17], horses [18,19,20,21,22] and dogs [10,11,23,24]. IRT detects infrared radiation, providing a pictorial representation of surface temperature [23]. Typically, vascular perfusion of the extremities changes during stress responses, which include increased arousal. In animals, this can be detected by a change in the surface temperature of superficial, hair-coat-free, anatomical landmarks of an animal that are perfused by extensive capillary networks, such as the eye and inside the ears [16]. Due to parasympathetic activation, dogs may exhibit an increase in heart rate and peripheral vasodilation upon the onset of a stress response, resulting in increased metabolic heat production, and an increase in surface temperature, which is most easily detected on the surface of the eye [23]. The benefit of selecting the eye for these temperature measurements is that it is unaffected by the length, moisture content and colour of any hair-coat at that location.

The current study aimed to determine possible effects of arousal and frustration on performance in racing greyhounds at race meets. Specifically, it was designed to explore putative behavioural indicators of increased arousal before racing and signs of frustration in the catching pen associated with being thwarted in capturing the lure. It involved obtaining IRT images of greyhounds before and after races, in parallel with behavioural observations of the greyhounds during the stir-up immediately prior to racing, and in the catching pen at the conclusion of races.

## 2. Materials and Methods

The University of Sydney Animal Ethics Committee approved the current study (Approval number: 2016/1015). The owners/handlers of the greyhounds provided informed consent for the collection of infrared images.

### 2.1. Location

The study was conducted at three greyhound racetracks in NSW over a period of 6 months. The tracks were Richmond and Wentworth Park in the Sydney metropolitan area in June and July 2017 respectively, and Gosford on the NSW Central Coast, approximately 80 km north of Sydney, in October and November 2017. Data were collected from 3 race meets at Richmond, with 11 races per meet, 2 race meets at Wentworth Park with 10 races per meet, and 3 race meets at Gosford with 8, 10 and 11 races, respectively. Race number for each race meet was recorded so that it could be used as an indicator of how long dogs had spent at the race meet before infrared images were taken immediately prior to their race.

Each track was configured differently (see Appendix A for diagrams). Minimum distances between features of the track and where on the grounds greyhounds were subject to potentially arousing stimuli were measured using the measurement tool in Google Earth Pro (Google Earth Pro version 7.3.2.5776, Google LLC 2019, Mountain View, CA, USA).

Races at all tracks are run over set distances. Wentworth Park races were 520 m or 720 m, Richmond races were 330 m, 400 m, 535 m or 618 m. Races at Gosford were 400 m, 515 m or 600 m. Dogs could compete over multiple distances across tracks. All tracks were sand tracks that were smoothed by a tractor between races. Catch pens at all tracks were soft sand to assist in safe deceleration and were not as well-lit as the racetrack, so after dark, dogs were running into dimmer lighting. All catch pens had opaque fencing at least up to greyhound head height, so dogs could not see through the fencing of the catch pen unless they stand on their hind legs.

Greyhounds were kennelled after vetting in temperature-controlled kennel blocks. It was permissible to leave them with a soft bed or blanket or to put a coat on them if the trainer/owner desires. They were given a small bowl of bottled water once kennelled. The kennels were large enough for a greyhound to lie down in and had opaque shared walls and wire front doors so that greyhounds could only see out of the front of the kennel and not have visual access to dogs kennelled on either side of them. Noise in the kennels could vary between tracks and race meets but consisted of barking, the sound of the lure once the race meet started, and the sounds of humans moving in and out of the kennel block and talking to one another. The racetrack’s public address system was also audible inside the kennel buildings.

During the period of data collection, the Richmond racetrack was trialling a bungee teaser in the catching pen. Teasers consisted of two toys made of synthetic fur attached to one bungee line each that was in turn anchored at the back fence of the catching pen. To offer teasers, the track steward operating the catching pen gate walked onto the track with the teasers, stretching both bungee lines taut. The steward released the teasers as the dogs approached the catching pen, providing a moving stimulus across the track and into the catching pen. The teasers came to rest in the sand trap of the catching pen (see Appendix A) and the dogs were able to interact with them. All greyhounds racing were muzzled, so interactions with the teaser were restricted. This system was in place for all race-meets where data were collected at Richmond.

### 2.2. Dogs

A total of 525 greyhounds were recruited to this study over the 8 race meets at 3 racetracks. The races included were for both male and female greyhounds aged 1–6 years old, and dogs varied in experience, with their number of starts ranging 0–177. The dogs arrived at the racetrack in air-conditioned dog trailers, which is the current policy of Greyhound Racing NSW (GRNSW). Upon entry to the racetrack they underwent vetting: an approximately 30-s veterinary physical examination to ensure the quality of health. The dogs were then kennelled in an air-conditioned building where they remained until they were taken out for pre-stir-up, approximately 10–15 min before their race. Dogs were excluded from data collection if they had previously been recorded by the current team of investigators at a prior race or race meet.

### 2.3. Physiological Data Collection

IRT data were collected twice from each dog during each race meet, with the first IR thermograph being taken during pre-stir-up. The second IR was taken 15 min after the race while the greyhound was kennelled. Post-race kennelling proceeds after greyhounds have been hosed down and offered water to drink upon finishing the race.

IRT images were captured using a FLIR T640 Professional Thermal Imaging camera (T640, FLIR Systems Inc. Danderyd, Sweden) at approximately a 100 angle from the human operator’s perspective (0° being straight up and 180° pointing at the operator’s feet) and 1 m distance from the dog.

Most race meets started at approximately 7 p.m., so when IRT images were taken there was abundant shade or the sun had set. Race meets at Gosford racetrack were visited in November when the sun set at approximately 7:30 p.m. IRT images at this track were mostly collected in an undercover area near the stir-up pen. This area had a north-westerly aspect and there was some direct sunlight in the stir-up pen itself for the first race of the evening. One race meet at Richmond started at 3:12 p.m. The sun set on this day at 4:58 p.m. The stir-up pen where IRT images were taken was not undercover but had a southerly aspect, so was shaded by nearby buildings.

The FLIR ResearchIR Max (v4.40, FLIR Systems Inc., Wilsonville, OR, USA) software program was used to calculate the average eye temperature under the 1234 palette because it best exposed the circumference of the eye. Greyhound eye temperature was calculated by tracing the eyelids of the greyhounds, using the Stats tool, then using Statistics Viewer to calculate the mean and maximum temperature inside the traced area.

### 2.4. Behavioural Data Collection

The behaviour of the dogs was recorded using one GoPro Hero3 Black Edition action camera (GoPro, Inc. San Mateo, CA, USA) mounted onto the fence of the catching pen, and one hand-held Sony HD Handycam HDR-PJ760 video camera (Sony Corporation, Sony City, Minato, Tokyo, Japan). The videos were analysed in slow motion in Windows Media Player 11 (Microsoft, Redmond, WA, USA) (0.5× speed) and counts of each of the behaviours listed in the ethogram (Table 1 and Table 2) were recorded for each dog. Greyhounds whose trainers excluded them from the optional stir-up event were not analysed with the ethogram and were recorded as being absent. Dogs were videoed in the stir-up from the moment the lure first started to move during the stir-up to the point at which all dogs had left the stir-up yard to walk to the start boxes. In the catching pen, video was started when the dogs entered the home straight and stopped when all dogs had been caught by handlers in the catching pen and were on a leash. All behaviours from the ethograms that occurred during filming were coded for analysis. Behaviours that would typically occur only once during the videoed period were recorded as only either yes/no, whereas behaviours that were likely to occur more than once during the videoed period were counted.

The ethograms were informed, in part, by Travain et. al. [23], who used an ethogram to estimate distress in a group of 14 dogs, and categorised behaviours as being indicative of distress when they were accompanied by a significant increase in eye temperature (detected by IRT) [23]. No racing-specific ethogram for dogs has been developed before so, for the current study, several behaviours were added to the ethogram if they were plausible candidates for detecting high arousal, frustration, or fixation on the lure.

### 2.5. Questionnaire Data Collection

The questionnaire for trainers consisted of 4 questions that were used to identify any indirect factors that may influence affective state:How long did it take you to get to the racetrack (minutes)?How many times has your greyhound raced (starts)?How long since the greyhound’s last race (days)?How old is your greyhound (years, rounding to the nearest half-year.)?

The data from this questionnaire were compiled along with the greyhounds’ start box number, the date of the race, track, race distance, performance (placing), time of the race meet, and ambient temperature at the time of the dog’s race. Ambient temperature was collected from records available from Time And Date AS (“Aksjeselskap”, Stavanger, Norway) which purchases weather information from customweather.com [25]. The records are available on an hourly basis for the Richmond, Sydney city, and Gosford localities.

### 2.6. Statistical Analysis

All statistical analyses were performed in RStudio (version 1.1.383, desktop macOS, RStudio Inc., Boston, MA, USA; R Foundation for Statistical Computing, Vienna, Austria). Behaviours were pooled into three categories to address some low counts. These categories were: “Aroused_S for behaviours” indicating arousal during stir-up, “Unresolved” for behaviours in the catching pen that may indicate the greyhound was still fixated on the unattainable lure or expressing frustration, and “Teaser” for behaviours in the catching pen at Richmond that were related to interacting with the teasers on bungee lines. The frequency of behaviour recordings both in the catching pen and in the stir-up were rarely more than 5 counts. The only exception was barking, which is energetically a much less costly behaviour than other behaviours in the ethograms and is also much quicker to perform. All behaviours were scaled using the max-min method to a scale of 0–5 counts to avoid the inflation of results in dogs prone to vocalisation. Counts for pooled behaviours were then rounded to the nearest whole number to allow for a negative binomial model to be fitted. This step was relevant only for Aroused_S behaviours, as behaviours in other categories did not need to be scaled. The model for Behaviours Indicating Arousal (BIA) included mean eye temperature before and after the race, race distance (Distance), racetrack (Track), sex, and Days_ last_race.

The final ordinal linear regression model for greyhound performance included the following factors: mean eye temperature before the race (Eye Temp Before); mean eye temperature after the race (Eye Temp After); ambient temperature (Temp); dog age (Age); start box number (Box); number of dogs in the race (Field); the number of days since the dog last raced (Days_last_race) and sex of the dog (Sex). It also included an interaction between Sex and Days_last_race.

Generalised linear models with a quasi-Poisson distribution due to over-dispersion in count data were used from the lme4 package using the glm function in RStudio to determine factors that had a significant effect on Aroused_S behaviours and Eye Temp Before races and Eye Temp After races. The generalised linear model for Eye Temp Before races contained the terms Track, Sex, Race and Aroused_S, and ambient temperature was included due to its presumed effect on surface temperature, although it worsened model fit. The generalised linear model for Eye Temp After races contained the terms Race, Temperature, Track, Distance, Sex, Aroused_S and Unresolved,

The final models were built using the stepwise method and the AIC number to determine the model of best fit. Pearson’s Correlation tests using the cor function were performed on factors that were not included in models or for which models were difficult to resolve.

## 3. Results

### 3.1. Tracks

Track configuration in terms of where the kennel block, stir-up yard and catching pen were located in relation to the track differed between tracks, as summarised in Table 3.

### 3.2. Performance

Increasing Eye Temp After had a negative association with performance (*n* = 290, Effect = −0.173, s.e. = 0.074, *p*-value = 0.027), and increasing age had a negative effect on performance (*n* = 290, Effect = −0.395, s.e. = 0.136, *p*-value = 0.004). On the whole, male dogs performed better than female dogs (*n* = 290, Effect = 0.752, s.e. = 0.257, *p*-value = 0.003), but they performed worse with increasing number of days since they were last raced, as shown in Figure 1 (*n* = 290, Effect = −0.022, s.e. = 0.010, *p*-value = 0.023). This is demonstrated further in Figure 2, which shows the predicted placings of male dogs for successive weeks since last raced when all other factors are held constant. These figures were obtained from the same ordinal model run on a subset of the original data containing only male dogs.

Performance for both sexes was influenced by the starting box number. Box 1 showed the strongest association with good performance while, in contrast, Boxes 4, 5 and 7 had a significantly negative effect on performance (see Table 4 for figures). Figure 3 shows the probability of placings from each starting box. The other factors did not have a statistically significant effect on performance, but their presence improved the model according to the AIC. The results of this model are shown in Table 3. Factors with a significant effect on performance appear in bold. Mean eye temperature 15-min after the race, Boxes 4, 5 and 7, increasing age, and days since last raced had a negative impact on performance for males only. Male dogs performed better than female dogs. The number of dogs in each race (field) was included in the model to account for the possible effects of there being fewer dogs in the race.

### 3.3. Behaviours Indicative of Arousal (BIA)

The most common BIA recorded in stir-up was lunging. The negative binomial model on the frequency of BIA was constructed in the same manner as the ordinal model. A summary of these results is shown in Table 5. Increasing race distance had a negative effect on the frequency of aroused behaviours in the stir-up (*n* = 290, Effect = -0.004, s.e. = 0.002, *p*-value = 0.031), and the race being held at Wentworth Park had a positive effect on the frequency of aroused behaviours in the stir-up (compared to Gosford) (*n* = 290, Effect = 1.255, s.e. = 0.380, *p*-value = 0.001).

### 3.4. Mean Eye Temperature

A summary of this model is shown in Table 6. Track had a powerful association with Eye Temp Before, with both Gosford and Wentworth having a strong, positive effect, as shown in Figure 4 (*n* = 442, Effect = 1.910, s.e. = 0.152, *p*-value = 0.001; Effect = 1.595, s.e. = 0.159, *p* < 0.001 for Gosford and Wentworth respectively). Increasing race number had a strong, positive effect on Eye Temp Before (*n* = 442, Effect = 0.103, s.e. = 0.022, *p*-value < 0.001), as shown in Figure 5. A generalised linear model for Eye Temp After races revealed that statistically significant predictors of Eye Temp After were ambient temperature (*n* = 310, Effect = 0.149, s.e. = 0.032, *p*-value < 0.001) and race number (an indicator of how long the dog has been at the race meet) had a positive effect (*n* = 310, Effect = 0.071, s.e = 0.027, *p* = 0.010) (Table 7). A scatter plot showing the relationship between ambient temperature and Mean Eye Temperature Before the race is shown in Figure 6. Mean Eye Temperature After the race was, as expected, positively influenced by ambient temperature.

There was a significant, negative correlation between Teaser-related behaviour and Eye Temp Before at Richmond track where the teasers were available in the catching pen (*n* = 166, correlation = −0.140, df = 446, *p*-value = 0.003). We were unable to resolve a model for the frequency of Unresolved behaviours in the catching pen or Teaser-related behaviours, which may be due to low count data combined with multiple factors having a small influence on these behaviours. However, the frequency of Unresolved behaviours in the catching pen at Richmond racetrack was dramatically lower (17.1% of starters) compared to Wentworth Park (77.1% of starters) and Gosford (96% of starters) racetracks.

## 4. Discussion

This study identified many factors that may contribute to greyhound performance at race meets, including age, sex, start box, and days since the dog last raced if male. Higher mean eye temperature indicative of heightened arousal or distress was influenced by what track the race meet was held at and the race number, which is a proxy for how long the dog has been kennelled.

### 4.1. Eye Temp after Races

Higher mean eye temperatures after the race were associated with poorer performance, but it is difficult to separate the effects of physical exertion in racing from the effects of emotional state before and during the races. Studies have found that eye temperature increases in response to physical exercise in dogs and horses [19,20,24,26], but the form of exercise in those studies was prolonged rather than the short intensity of sprint races in the current study. The current mean eye temperatures after racing may be indicative of a stronger stress response to racing, and the disruptive effects of over-arousal on performance. An alternative reason for this relationship is that increased mean eye temperature after racing is more indicative of higher core body temperature than directly of emotional state. A previous study suggested it takes at least 30-min for dogs’ core body temperature to return to baseline after 30-min of exercise [24]. In the current study, it was not possible to collect IRT images more than 15-min after racing at the race meets due to owners removing them from kennels. As such, the negative relationship between performance and observed mean eye temperatures after the race may represent dogs that perform poorly having to invest more effort to compete in the race than dogs that perform well, and thus having higher core body temperatures and mean eye temperatures after the race.

Sampling in the kennels prior to racing would likely provide a better comparison of before and after racing as the act of taking the dogs out of the kennels may elevate their arousal. However, for the current study, this was not permitted by the racing officials. Under the rules of racing [4], greyhounds must remain in the kennels for 15 min before they can be taken home, so the IR thermograph needed to be taken at this 15-min juncture to allow the greyhounds to cool down so as to minimise the effects of core body temperature on eye surface temperature, while still obtaining thermographs before the greyhounds were removed from their kennels to trailers for transportation home.

### 4.2. Ambient Temperature

Ambient temperature was a significant predictor of mean eye temperatures after the race, but not of mean eye temperatures before the race. This variable reduced the goodness of model fit, which may reflect a non-linear effect of ambient temperature. Greyhounds are transported in air-conditioned trailers and held in temperature-controlled kennels before their race, but extreme ambient temperatures may influence IRT images after races, and this may need to be treated with caution if the industry elects to use IRT-based monitoring of dogs in future. Likewise, direct sunlight may overshadow any variability in surface temperature as measured by IRT. Further research into eye temperatures after standardised highly arousing activities and various intensities of physical exercise will reveal the utility and limitations of IR images in assessing emotional states in dogs of various breeds and levels of fitness.

### 4.3. Race Number

There was a significant positive effect of race number on mean eye temperatures before the race, suggesting that greyhounds at the race meet grew increasingly aroused as the race meet progressed. All greyhounds racing must be kennelled 30 min before the first race. They are undisturbed in the period between kennelling closing and the first race, but once races are underway, a steady stream of handlers enter the kennels to collect dogs and return dogs that have just raced. The kennelled dogs are therefore exposed to ongoing disturbance, and also likely hear arousing auditory stimuli from outside the kennel area, such as the sound of the lure moving on the track. This effect is unlikely to be related to the intensity of competition, as prize money does not routinely increase with each race. Greyhounds cannot know when they will be taken out of their kennel for racing, so may anticipate this occurring every time they are disturbed by the movements of trainers and dogs, leading to increasing frustration and aroused anticipation when they are not removed from the kennel.

### 4.4. Behaviours Indicative of Arousal (BIA) in Stir-Up

A previous study found that horses displaying BIA immediately prior to racing performed more poorly than horses that appeared calmer [6]. The current study found no such a relationship between performance in the racing greyhound and either mean eye temperatures before the race or the frequency of aroused behaviours in the stir-up. Indeed, there were no relationships among the behaviours in the current ethogram thought to indicate emotional arousal, mean eye temperature before or after the race, and performance. Aroused behaviours in stir-up were best explained by race distance and race venue, with fewer behaviours indicative of arousal being observed before longer race distances, and more of such behaviours at the Wentworth Park racetrack than at the other tracks. Greyhounds can be run over multiple distances across various tracks, but there can be specialisation. Greyhounds in the stir-up have no way to know what distance the ensuing race will be, but it is possible dogs take cues from their handlers and that handlers do not encourage lively responses in the stir-up for longer races. It is also possible that dogs that are less active in stir-up are better suited to longer races, and so fewer observable BIA may arise in the stir-ups before longer races.

These results call into question both the validity of ethograms in assessing arousal at race meetings, and whether greyhounds encounter elevated arousal minutes before the mandatory stir-up. It was not feasible to collect IRT images of greyhounds during or immediately following stir-up due to the pressing schedules of race meetings, so it is possible many greyhounds were not experiencing heightened arousal when the pre-race IRT images were being taken. Alternatively, the behaviour of dogs during stir-up may be a poor indicator of arousal. Dogs that express their arousal overtly with easily detectable behaviours may be no more aroused than those that are passive during stir-up. Any difference in behaviours may not directly reflect a difference in arousal level.

### 4.5. Age and Experience

Younger dogs were more likely to place in the front half of the field than older dogs. This may be because they are less likely to be burdened with the effects of previous injuries or general degenerative changes that may occur with age. In racehorses, studies have found that the risk of injury increases with age [27,28] and that racing speeds peak [29] or plateau [30] at approximately 4.5 years of age.

### 4.6. Start Box

The box from which dogs started also had a significant impact on their likelihood of placing favourably, an outcome which is freely acknowledged by track administrators, for Wentworth Park at least [31]. Box 1 appears to offer an advantage, as noted by The Greyhound Recorder [32], and in the current study, Boxes 4, 5 and 7 conferred a significant disadvantage when compared to Box 1. Greyhounds that prefer to run close to the rail are likely to perform better regardless of starting box because they must cover less ground over the course of the race than greyhounds that prefer to run on the outside of the field. As such, greyhounds that prefer to run close to the rail and that also start from Box 1, 2 or 3 are likely to cover less ground than greyhounds with this preference that start from boxes farther from the rail. This issue may be best addressed by adopting track safety recommendations for a lure system that places the lure closer to the centre of the track [33].

### 4.7. Sex of Dog

Male greyhounds in the current study were significantly more likely to place favourably than females. However, this was complicated by an interaction between sex and days since last raced. Whereas females showed no clear pattern in their performance regardless of how long it had been since they were last raced, males were more likely to place poorly the longer it had been since they last raced. There was no significant difference between males and females in latency since the previous race. The effect of this interaction on performance is intriguing but small and difficult to interpret. There was a significant negative correlation between mean eye temperatures before the races and latency since the previous race. This is at odds with a potential explanation of a more intense stress response to the race meet environment after longer rest periods. It is possible that increasing latency since the previous race compromises musculoskeletal strength and race fitness, as extant data on racehorses show an increase in the likelihood of sustaining a serious injury during a race with increasing days since last racing [34]. It is also possible that anticipation of racing is diminished by the lack of recent associations with track-related stimuli if the dog has not raced for more than a week or so, and that this compromises performance. This is consistent with the current finding of a negative correlation with mean eye temperatures before the races.

Why latency since the previous race should affect male performance more than female performance is unclear and, to the authors’ knowledge, has not previously been reported in animal performance studies. As always, it is possible this reflects a statistical anomaly, and that simply increasing the sample size would clarify the strength of this relationship.

There was no significant difference in mean eye temperatures between sexes before or after the race and, to the authors’ knowledge, no sex differences in response to arousal have previously been reported in dogs.

### 4.8. Catching Pens

One of the goals of the current study was to investigate whether greyhounds in NSW races were being sufficiently rewarded for racing despite being unable to access the lure at the conclusion of races. We may assume that if greyhounds are finishing races without any penalty for failure to chase the lure, they are being sufficiently reinforced for racing at the time of observing them race. However, it is possible that reinforcement will not be sufficient to maintain racing over time. As such, we searched for signs of frustration in the catching pen upon conclusion of the race. Frustration has been associated with increased aggression in dogs [35,36,37], which in turn, may manifest as an increased risk of attracting a penalty for marring during the race. It is also likely that what occurs in the catching pen influences a greyhound’s emotional associations with racing in general, and their willingness to enter the catching pen. For the purposes of the current study, the putative behavioural signs of frustration in the catching pen included jostling another dog, focusing on the lure gate, and changing directions (included only if the dog initiated the direction change rather than following another dog that had changed direction). We found these behaviours in 59.1% of greyhounds.

The prevalence of these behaviours is concerning for two reasons. Firstly, it suggests that, at the end of the race, many greyhounds are still focused on chasing the lure when that opportunity is taken from them. If they are going to change direction, greyhounds almost always do so along the inside fence of the track or catching pen closest to the path of the lure arm, and often do so by orienting the head towards the lure’s direction of travel. The lure in transit makes a loud noise and, after passing the catching pen, may traverse more than half the track before coming to a halt, so the greyhounds can still hear it while in the catching pen, unable to chase it. The greyhounds orienting towards the lure gate indicates their focus on where they last saw the lure.

The jostling that occurred after nearly every race in the current study is not necessarily an aggressive expression of frustration. However, the redirection of aggression towards conspecifics is a recognised consequence of being thwarted in obtaining a goal (see Reference [38]). Other possible causes of this behaviour may be that it is playful in nature, or it may be a product of up to eight galloping dogs coming to a halt together in a relatively small space. The dogs behind those in the lead may take a moment to react to the deceleration of dogs in front of them, resulting in bunching. Nonetheless, the indications that the greyhounds are still fixated on the lure raise the prospect that they are not disengaging from their goal even though they are unable to continue pursuing it. If their goal is to capture the lure, it may be less important whether they are successful or not and more important that they are not consistently prevented from pursuing the lure when they can still hear it. The effects of frustration on behaviour may include heightened arousal as well as depressive disappointment [38]. Even if only a small number of greyhounds encounter the most detrimental effects of frustration in the catching pen, they may be subject to reduced interest in pursuing the rewards in question; an established response to frustrated non-reward [39].

Only 11.3% of greyhounds at Richmond track were observed showing direct interest in the teasers in the catching pen. In contrast, 59.1% of greyhounds across all racetracks showed unresolved behaviours in the catching pen, and this was more than three times higher at the Wentworth Park and Gosford racetracks than at the Richmond racetrack. The distinctly lower incidence of unresolved behaviours in the catching pen at Richmond racetrack may arise from the use of teasers in the catching pen, even though only a small portion of the racing greyhound population racing at Richmond showed an active interest in the teasers.

### 4.9. Track Effects

There was a significant difference among mean eye temperatures, both before and after races, at different tracks. This between-track difference was strongest among eye temperature before races, which is the metric most directly reflective of stressors arising in the general racetrack environment. A general linear model, with mean eye temperatures before races as the independent variable, revealed a significant positive influence of both Wentworth Park and Gosford compared to Richmond, with Richmond being associated with the lowest mean eye temperatures before the race. This may have been influenced by ambient temperature, but the effect of ambient temperature was neither strong nor significant. This finding suggests that some attributes of the Gosford racetrack may be inherently more stressful than Wentworth Park and that those of Richmond may be less stressful than those of Wentworth Park. Attributes worth considering include the design of the kennel facilities, as the catch pen, stir-up and kennel block are all very close to each other at Wentworth Park, but more separated at Gosford and at Richmond the kennel block is farthest of all three tracks from both the racing track (see Appendix A). The operational behaviour of personnel at the track and how handlers manage and interact with the dogs, or how quickly they enter the catching pen and restrain the dogs may also have an impact, or it may be that these tracks are particularly attractive to owners of greyhounds that are more or less prone to distress. There were also significantly more behaviours indicative of arousal in the stir-up at Wentworth Park than at Gosford. This may reflect the effects of a track design factor or an operational factor.

The Richmond difference may also, to some extent, be attributable to the presence of teasers in the catching pen. Further investigation into which track attributes may influence greyhound stress before racing is important for the integrity of the sport so that vetting, kennelling, and pre-stir-up procedures can be designed to support greyhounds equitably.

## 5. Conclusions

The need to understand behavioural wastage in racing greyhounds is clear. Age, start box, sex of dog, spelling and track effects all affect the performance of greyhounds on tracks.

This is the first published study of racing greyhound behaviour at race meets and the first, for any racing code, to use IRT at racetracks to assess arousal. The results describe modest relationships between eye temperature and performance. This may assist in the development of more detailed studies to identify specific factors that compromise performance and establish how they can be modified to reduce their negative effects. IRT before races may be more revealing than IRT after races due to the influence of core body temperature that may reflect the legacy effects of physical effort more than arousal, and the influence of ambient temperature. Attempts to use IRT before races bring significant timing and logistical challenges, but this study showed there is promise in eye temperature measurements before races to reveal the effects of greyhounds’ experience at racetracks in the long- and short-term on behaviour and performance. For clearer results, it would be worthwhile investigating how IRT images may be collected closer in time to the exertion of racing.

This study also offers insights into how individuals within the racing greyhound population respond differently to the anticipation of racing. The use of teasers in the catching pen and the track effects on behaviour and arousal are both areas that merit further research to make race-meets optimally arousing for racing greyhounds, and to improve reward availability and appropriate disengagement from the lure at the end of races. The large percentage of dogs showing signs of frustration and continuing to search for the lure when in the catching pen raises concern for both the physical and emotional wellbeing of greyhounds. The catching pen system has been in operation for many years but may not support all racing greyhounds equitably in avoiding wastage. The period during which greyhounds are kennelled before their race may contribute to their stress at race meets, so any means by which kennelling time can be reduced, especially for greyhounds in races late in each meet’s program, are worth further investigation and suitable trials.

## Figures and Tables

**Figure 1 animals-10-01037-f001:**
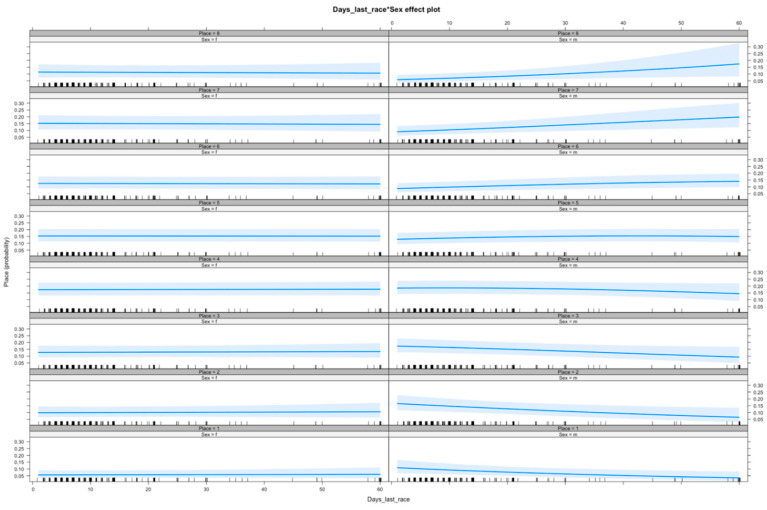
The effects of days since last raced (x-axis) on the probability of placing (place 1st–8th) for females (left column) and males (right column). There was an interaction between sex and days since last raced, with males more likely to place poorly as intervals since last racing increased. Confidence intervals (95%) shown in shading.

**Figure 2 animals-10-01037-f002:**
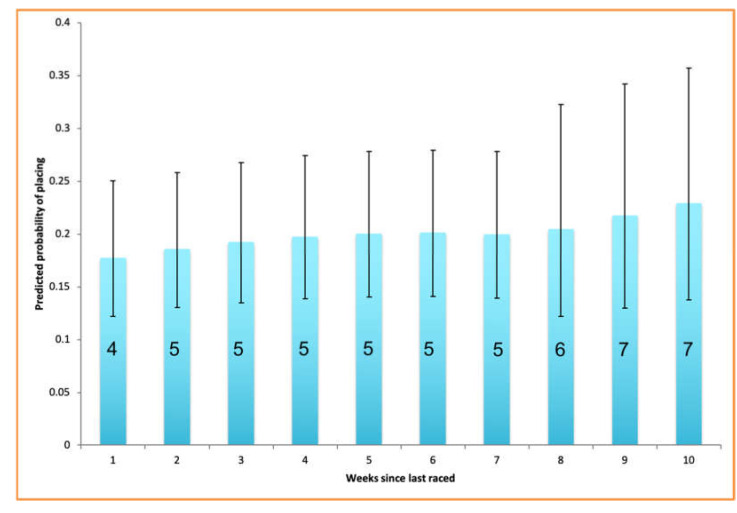
Predicted placings of male dogs 1–10 weeks after since last racing. Placings with the highest probability when all other factors in the ordinal linear regression model are held at their mean are shown on the bars. Error bars show the upper limits and lower limits of predicted probabilities associated with the placings. Thus, there is a predicted loss of three places between racing male dogs a week after their last race compared to 10 weeks after their last race.

**Figure 3 animals-10-01037-f003:**
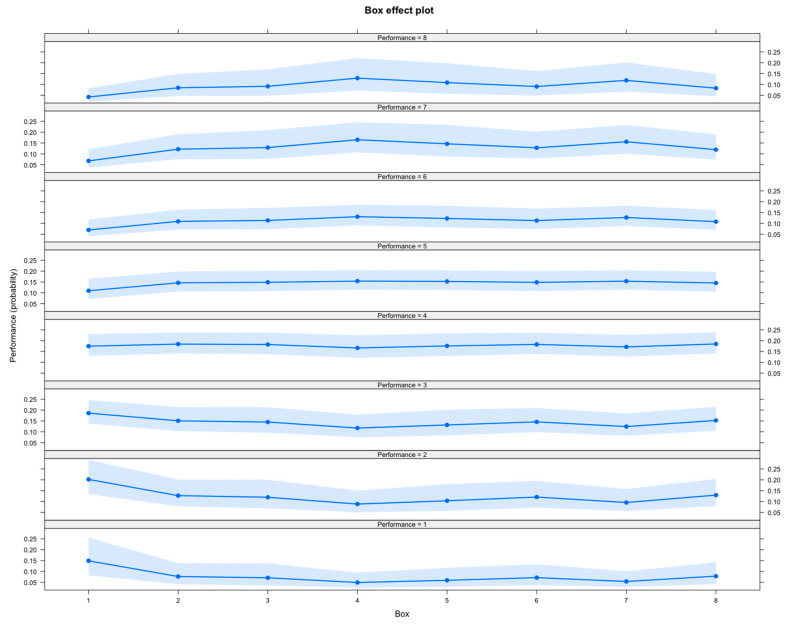
The effects of the starting box on the probability of placing 1st–8th (performance—one graph for each placing). Dogs have a higher probability of placing first or second if they start from Box 1. Boxes 4 and 5 are associated with an elevated probability of placing 7th or 8th. Confidence intervals (95%) are shown in shading. Box is on the x-axis and probability given placing (performance) is on the y-axis.

**Figure 4 animals-10-01037-f004:**
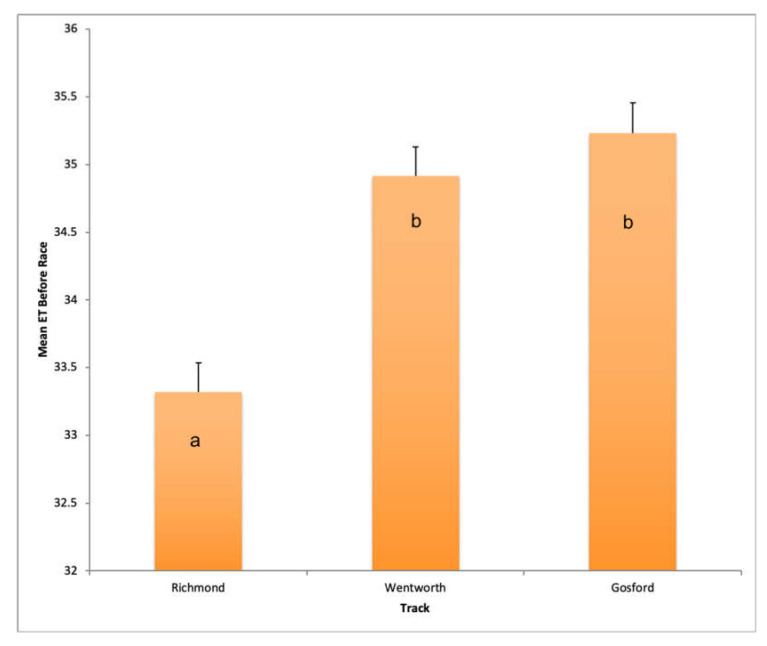
Fitted Eye Temp Before races at different tracks. Eye Temp Before was much lower at Richmond than the other two tracks, suggesting dogs are overall calmer at Richmond racetrack than Wentworth Park or Gosford. ^a,b^ Superscripts are assigned to values that are significantly different from one another.

**Figure 5 animals-10-01037-f005:**
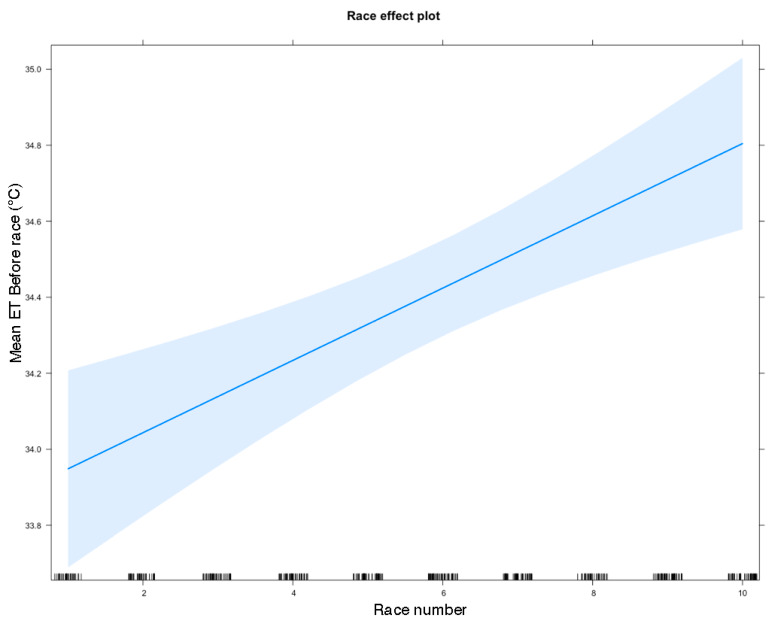
Predicted Eye Temp Before races depending on race number. Eye Temp Before races increases as race number increases, indicating greyhounds become increasingly aroused as the race meet progresses. 95% confidence intervals shown with shading. Ticks on the x-axis give an indication of n for each race number.

**Figure 6 animals-10-01037-f006:**
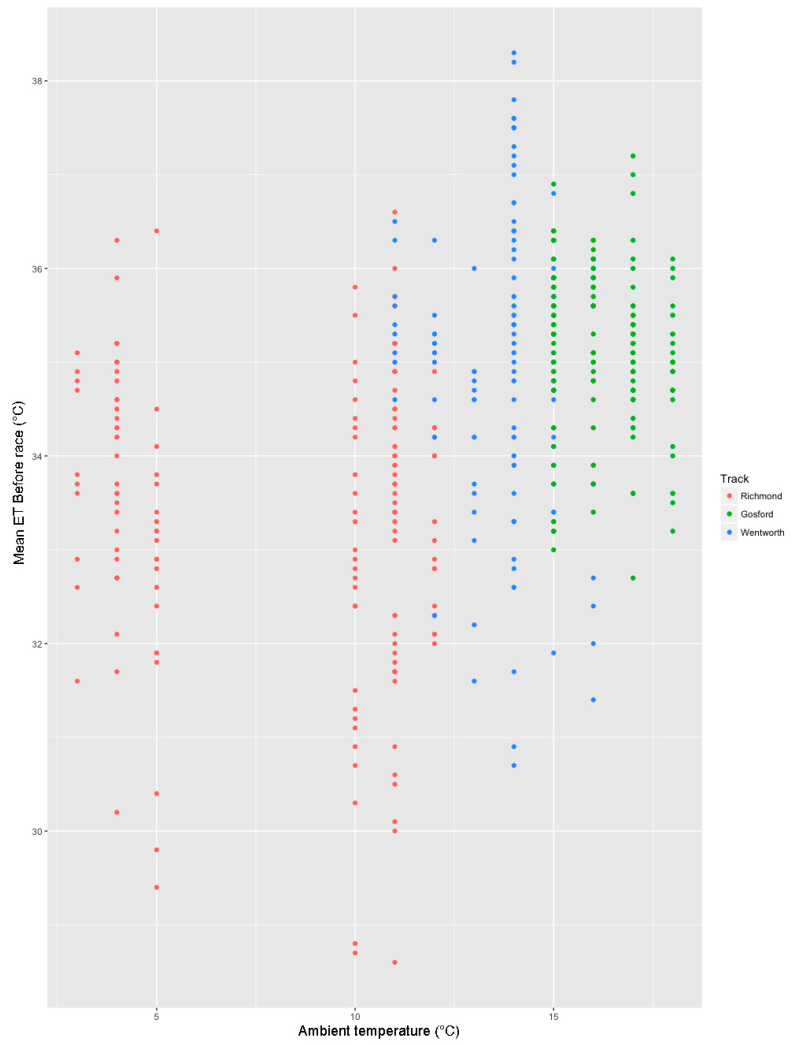
Scatter plot showing the relationship between Eye Temp Before the race and ambient temperature for each track. Ambient temperature was not included in the model to predict Eye Temp Before because its addition to the model worsened the fit of the model according to the Akaike Information Criterion (AIC).

**Table 1 animals-10-01037-t001:** Ethogram of all the behaviours potentially indicating arousal during stir-up in greyhounds.

Behaviour	Description	Frequency
Rising	Unassisted rising onto hind legs without hind feet leaving the ground in vertical or lateral movement	Count
Owner-assisted rising	Owner lifting dog onto hind legs without hind feet leaving the ground in vertical or lateral movement	Count
Lunging	Lateral thrust forward, therefore pulling on the leash	Count
Spinning	Dog rotates laterally either clockwise or counter-clockwise for approximately a full revolution	Count
Jumping	Both front and back feet leaving the ground so that a suspension phase occurs	Count
Barking	Barking	Count

**Table 2 animals-10-01037-t002:** Ethogram of all the behaviours that potentially indicate negative emotional valence in greyhounds in the catching pen after a race.

Behaviour	Description	Frequency
Grabbing the teaser	Teeth contact the teaser but are released before trainers contact the dog	Yes/no
Changing directions	An approximate 180° change in direction while in motion.	Count
Jostling	Dog’s muzzle makes physical contact with another dog with sufficient force to affect the receiving dog	Count
Attention directed to lure gate on the racetrack	Dog orientating body position and interest towards the lure gate, or attention to the lure as it slows down around the track	Yes/no
Holding teaser	Teaser is grabbed and not released by the greyhound	Yes/no

**Table 3 animals-10-01037-t003:** Distances in metres between features of each racetrack in the study.

Track	Kennels to Stir-Up (m)	Kennels to Track (m)	Stir-Up to Track (m)	Catching Pen to Kennels (m)	Catching Pen to Stir-Up (m)
Richmond	5	34	18	140	123
Wentworth Park	2	8	1	5	2
Gosford	17	28	3	170	150

**Table 4 animals-10-01037-t004:** Summary of an ordinal linear regression model on performance in racing greyhounds.

Term	Effect	S.E.	*p*-Value
Eye Temp Before	0.011	0.068	0.885
Eye Temp After	−0.173	0.074	0.027 *
Sex (m)	0.752	0.257	0.003 *
Box2	−0.746	0.417	0.073
Box3	−0.830	0.447	0.063
Box4	−1.220	0.425	0.004 *
Box5	−1.023	0.455	0.025 *
Box6	−0.820	0.425	0.053
Box7	−1.122	0.418	0.007 *
Box8	−0.723	0.413	0.080
Age	−0.395	0.136	0.004 *
Days_last_race	0.001	0.005	0.791
Field	−0.459	0.177	0.010 *
Temp	0.002	0.029	0.954
Sexm:Days_last_race	−0.022	0.010	0.023 *

* Statistically significant (*p* < 0.05) effects. S.E = standard error.

**Table 5 animals-10-01037-t005:** Summary of a negative binomial linear regression model of BIA during stir-up (immediately before racing). Increasing race distance had a negative effect on the frequency of aroused behaviours. More aroused behaviours were observed at the Wentworth Park track than at the Gosford track. S.E. = standard error.

Term	Effect	S.E.	*p*-Value
(Intercept)	−0.204	4.733	0.966
Eye Temp Before	−0.052	0.092	0.574
Eye Temp After	0.094	0.093	0.310
Sex (m)	0.198	0.263	0.450
Distance	−0.004	0.002	0.031 *
Track (Richmond)	0.587	0.400	0.142
Track (Wentworth)	1.255	0.380	0.001 *
Days_last_race	−0.015	0.008	0.071

* Statistically significant results.

**Table 6 animals-10-01037-t006:** Generalised linear model summary for Eye Temp Before races. Track and increasing race number both have significant, positive effects on Eye Temp Before races.

Term	Effect	S.E.	*p*-Value
(Intercept)	32.676	0.328	<0.001
Race	0.103	0.002	<0.001
Track (Gosford)	1.910	0.274	<0.001
Track (Wentworth)	1.595	0.221	<0.001
Sexm	−0.003	0.130	0.983
Aroused_S	0.006	0.034	0.850
Temp	0.027	0.029	0.361

**Table 7 animals-10-01037-t007:** Generalised linear model summary for Eye Temp After races. Ambient temperature (Celsius) and increasing race number both have significant, positive effects on Eye Temp After races. S.E. = standard error.

Term	Estimate	S.E.	*p*-Value
(Intercept)	33.416	0.704	<0.001 *
Race	0.071	0.027	0.01 *
Distance	−0.001	0.001	0.258
Sex (m)	−0.014	0.157	0.927
Track (Gosford)	0.29	0.346	0.402
Track (Wentworth)	−0.05	0.325	0.878
Aroused	0.04	0.041	0.328
Unresolved	0.055	0.136	0.688
Temperature	0.159	0.032	<0.001 *

* Statistically significant effects.

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
