# Peer review of "A Pilot Study of Methods for Evaluating the Effects of Arousal and Emotional Valence on Performance of Racing Greyhounds"

_animals, 2020, doi:10.3390/ani10061037_

Round 1

Reviewer 1 Report

The manuscript "A pilot study on methods of evaluating the effects of around and emotional valence on performance of racing greyhounds" describes an important topic of greyhound racing industry. Dogs bred only for racing purposes must have specific characteristics to be able to perform that role and the most important one is desire for chasing. As specified in the study, failure to chase is considered a form of behavioral wastage, which is where otherwise physically healthy and sound animals are removed from a role because they are unable to perform it adequately due to behavioral unsuitability. Such dogs are eliminated from races and often euthanized if they do not find a home.

The research group of 525 greyhound, on which the observations were made, is representative.
Separation of factors affecting the stimulation of dogs is logical and sufficient (Mean ET After Races, Ambient temperaturÄ™, Race numer, Behaviours indicative of arousal in stir-up, Age and experience, Start box, Sex of dog, Catching pens, Track effects).

The risk of misinterpretation of the behavior of kenneled dogs before and after the race, exposed to ongoing disturbance, and seeing other dogs enter and leave, was related to the rules of dog racing and the authors had no influence on them (line 356-361 and 378-384).

It would be good to complete the paper by elaborating on what stimulation behaviors occur most often.
The reduction of stress factors on racing tracks contributes to the improvement of welfare and to longer use of racing dogs. This is why it is important to determine the factors causing unnecessary emotional arousal in dogs.
The manuscript can be published in the journal Animals after minor revision

Author Response

The authors are unclear what “stimulation behaviours” refers to. We have added the following in the results section under BIA.

The most common BIA recorded in stir-up was lunging.

Reviewer 2 Report

The study "A pilot study on methods of evaluating the effects of arousal and emotional valence on performance of racing greyhounds" investigated the relationship of physiological measures (eye temperature), arousal behaviors, and track factors on performance in greyhounds. The topic is an important one in the aim of reducing wastage and improving welfare. 

The study was conducted well and my comments are mainly to help clarify some of those methods or analyses. 

--Behavioural data collection. Could you indicate how long you video recorded the dogs in the catching pen and whether you scored that entire video or only certain time points?

--Figure 2. Please clarify what the numerals are in the bars of the bar graph. In Figure 2 and Figure 4, remove background horizontal bars. Maximize the data:ink ratio.

--In the tables, the captions indicated statistically significant results were bolded but none were bolded. This might be a translation issue when producing the PDF for review but please check that. There was a stray bolded "a" in the word "appear" in the caption for Table 4. 

--Figure 5. Please explain the bottom vertical ticks. I took these as a count but it would be helpful to be explicit in the caption. Also, since there are only 10 values, have all values on the x-axis.

--Please explain the effect of race distance. It might help to elaborate that it seems the dogs are raced repeatedly at the same track. So, is this the dog anticipating the distance because this is the only track it races at or is it because it's been raced several times that day already and the amount of racing the dog has done has affected the BIA?

--Up until I read the discussion, I hadn't realized you had collected the race number (when I read the results I assumed that was the # of starts the dog had made). Please clarify race number in your methods and the results, including whether this was a track factor (for example the track had 15 races in a day) or a dog factor (the data were collected from the dog when the dog had run in the third race run at the track that day).

--L445-446 were unclear. 

--A few typos (there was a "euthanased" on line 55) and some of the in-text citations did not come through as numbers. 

Author Response

The study "A pilot study on methods of evaluating the effects of arousal and emotional valence on performance of racing greyhounds" investigated the relationship of physiological measures (eye temperature), arousal behaviors, and track factors on performance in greyhounds. The topic is an important one in the aim of reducing wastage and improving welfare. 

The study was conducted well and my comments are mainly to help clarify some of those methods or analyses. 

--Behavioural data collection. Could you indicate how long you video recorded the dogs in the catching pen and whether you scored that entire video or only certain time points?

The following was added to the manuscript:

Dogs were videoed in the stir-up from the moment the lure first started to move during the stir-up to the point at which all dogs had left the stir-up yard to walk to the start boxes. In the catch pen, video was started when the dogs entered the home straight and stopped when all dogs had been caught by handlers in the catch pen and were on leash. All behaviours from the ethograms that occurred during filming were coded for analysis.

--Figure 2. Please clarify what the numerals are in the bars of the bar graph. In Figure 2 and Figure 4, remove background horizontal bars. Maximize the data:ink ratio.

Replaced “Place” with “Placings” in the caption of Figure 2 to clarify. Removed horizontal lines in both Figures.

--In the tables, the captions indicated statistically significant results were bolded but none were bolded. This might be a translation issue when producing the PDF for review but please check that. There was a stray bolded "a" in the word "appear" in the caption for Table 4. 

Tables have been re-checked and bold text is now consistent with captions.

--Figure 5. Please explain the bottom vertical ticks. I took these as a count but it would be helpful to be explicit in the caption. Also, since there are only 10 values, have all values on the x-axis.

Added to caption: Ticks on x axis give an indication of n for each race number.

--Please explain the effect of race distance. It might help to elaborate that it seems the dogs are raced repeatedly at the same track. So, is this the dog anticipating the distance because this is the only track it races at or is it because it's been raced several times that day already and the amount of racing the dog has done has affected the BIA?

Dogs are raced repeatedly at the same track, but not usually in the same race meeting. However, dogs often specialise in distances. They may run specific distances or they may run varied distances. The following text has been added to the methods section to clarify this:

Races at all tracks are run over set distances. Wentworth Park races were 520m or 720m, Richmond races were 330m, 400m, 535m or 618m. Races at Gosford were 400m, 515m or 600m. Dogs could compete over multiple distances

The following text was added to the BIA section in the Discussion:

Greyhounds can be run over multiple distances across various tracks, but there can be specialisation. Greyhounds in the stir-up have no way to know what distance the ensuing race will be, but it is possible dogs take cues from their handlers and that handlers do not encourage lively responses in the stir-up for longer races. It is also possible that dogs that are less active in stir-up are better suited to longer races, and so there  fewer observable BIA may arise in the stir-ups before longer races.

--Up until I read the discussion, I hadn't realized you had collected the race number (when I read the results I assumed that was the # of starts the dog had made). Please clarify race number in your methods and the results, including whether this was a track factor (for example the track had 15 races in a day) or a dog factor (the data were collected from the dog when the dog had run in the third race run at the track that day).

Added to methods:

Race number for each race meet was recorded so that it could be used as an indicator of how long dogs had spent at the race meet before infrared images were taken, immediately prior to their race

Additional text added to results:

…and race number (an indicator of how long the dog has been at the race meet) had a positive effect.

--L445-446 were unclear. 

The sentence indicated is: However, recording one race per greyhound cannot demonstrate that greyhounds are being sufficiently rewarded to continue racing indefinitely.

It has been changed to read:

However, it is possible that reinforcement will not be sufficient to maintain racing over time. As such, we searched for signs of frustration in the catching pen upon conclusion of the race.

--A few typos (there was a "euthanased" on line 55) and some of the in-text citations did not come through as numbers. 

Corrected, thank you.

Reviewer 3 Report

I found this manuscript a pleasure to read, clearly well thought out and organized. I have a few comments.

Lines 30,31 – Not sure if this is a typo here, it starts with a “=” and “Increasing MeanETAfter” not sure what that means (perhaps from your model but needs to be renamed?)

Lines 61,62 – Some errors in your insertion of citations

Table 2 – What is the difference between “yes/no” and “count”? Were “yes/no” behaviors only counted one time? Did the dogs only perform the behaviors one time? I am confused why these behaviors weren’t also a count, please explain.

Line 210 – It is not sufficient to cite only RStudio, as it is just a dashboard for R – please cite R also

Line 219 – Is there a citation for this method to ensure methods are repeatable?

Line 236 – missing a period

Results generally – I think it might be helpful to rename or at least redefine your variables here, since the names you gave them are not themselves descriptive (I had to keep scrolling up to check what exactly MeanETAxx was, for example).

Line 272 – Did all of these things have a negative affect on only male performance, or only “days since last race”? Please clarify, as this sentence reads as if it is all of those things, but the model suggests only a sex by days since last race interaction

Discussion – In general I think the conclusions are well supported, but I am not sure I am sold on the main reason for the generally lower stress (as measured by your study) is only from the extra lure toy in the pen at the Richmond track. For example, Table 3 shows that the stir-up to track is significantly longer at this track, perhaps the short “cooldown walk” is resulting in this lower stress as well? Particularly considering the vast majority of the dogs did not interact with the toy. Similarly, at Wentworth, where you saw a high amount of stress, the catching pen to kennels is significantly lower than the other two tracks (actually, significantly lower in ALL distances). I think it might be something to consider, if you had not already (and I just missed it). It would not be possible to put it in the models, since it varies necessarily by 'track' which you already have, but there is data out there discussion cool down walks on dog/animal stress, that would be pertinent to add to your discussion for future studies, given these vast differences in track layouts.

Supplemental Material – I assume it just was not uploaded, but there is no discussion of what A, B or C refer to.

Author Response

Lines 30,31 – Not sure if this is a there, it starts with a “=” and “Increasing MeanETAfter” not sure what that means (perhaps from your model but needs to be renamed?)

The “=” sign has been removed, thank you.

Lines 61,62 – Some errors in your insertion of citations

Thank you for picking this up. It has been corrected.

Table 2 – What is the difference between “yes/no” and “count”? Were “yes/no” behaviors only counted one time? Did the dogs only perform the behaviors one time? I am confused why these behaviors weren’t also a count, please explain.

The following text has been inserted to explain:

Behaviours that would typically occur only once during the videoed period were recorded as only either yes/no, whereas behaviours that were likely to occur more than once during the videoed period were counted.

Line 210 – It is not sufficient to cite only RStudio, as it is just a dashboard for R – please cite R also

Thank you, this has been corrected.

Line 219 – Is there a citation for this method to ensure methods are repeatable?

The methods can be cited from this manuscript if they are repeated.

Line 236 – missing a period

Corrected, thank you.

Results generally – I think it might be helpful to rename or at least redefine your variables here, since the names you gave them are not themselves descriptive (I had to keep scrolling up to check what exactly MeanETAxx was, for example).

MeanETBefore has been changed to Eye Temp Before and MeanETAfter has been changed to Eye Temp After.

Line 272 – Did all of these things have a negative affect on only male performance, or only “days since last race”? Please clarify, as this sentence reads as if it is all of those things, but the model suggests only a sex by days since last race interaction

Only “days since last race”. The text has been adjusted so the paragraph starts with the following sentence:

Performance for both sexes was influenced by the starting box number.

Discussion – In general I think the conclusions are well supported, but I am not sure I am sold on the main reason for the generally lower stress (as measured by your study) is only from the extra lure toy in the pen at the Richmond track. For example, Table 3 shows that the stir-up to track is significantly longer at this track, perhaps the short “cooldown walk” is resulting in this lower stress as well? Particularly considering the vast majority of the dogs did not interact with the toy. Similarly, at Wentworth, where you saw a high amount of stress, the catching pen to kennels is significantly lower than the other two tracks (actually, significantly lower in ALL distances). I think it might be something to consider, if you had not already (and I just missed it). It would not be possible to put it in the models, since it varies necessarily by 'track' which you already have, but there is data out there discussion cool down walks on dog/animal stress, that would be pertinent to add to your discussion for future studies, given these vast differences in track layouts.

The section on track effects in the Discussion only contains one sentence on the presence of the teaser toy. It was unintentional for this to give the impression it is considered the main difference between the tracks. The following text has been added to this section to focus it on track configuration:

Attributes worth considering include the design of the kennel facilities, because the catch pen, stir-up and kennel block are all very close to each other at Wentworth Park, but are more separated at Gosford while at Richmond the kennel block is farthest (of all three tracks) from the racetrack (see Supplemental Material).

Supplemental Material – I assume it just was not uploaded, but there is no discussion of what A, B or C refer to.

The caption for the SM is on line 604: Supplementary Materials: The following are available online at www.mdpi.com/xxx/s1, Figure S1: Configurations of the three tracks in the study. a) Kennel block; b) stir-up yard; c) catch-pen.